# Artificial Intelligence in Medical Imaging: Analyzing the Performance of ChatGPT and Microsoft Bing in Scoliosis Detection and Cobb Angle Assessment

**DOI:** 10.3390/diagnostics14070773

**Published:** 2024-04-05

**Authors:** Artur Fabijan, Agnieszka Zawadzka-Fabijan, Robert Fabijan, Krzysztof Zakrzewski, Emilia Nowosławska, Bartosz Polis

**Affiliations:** 1Department of Neurosurgery, Polish-Mother’s Memorial Hospital Research Institute, 93-338 Lodz, Poland; krzysztof.zakrzewski@iczmp.edu.pl (K.Z.); emilia.nowoslawska@iczmp.edu.pl (E.N.); jezza@post.pl (B.P.); 2Department of Rehabilitation Medicine, Faculty of Health Sciences, Medical University of Lodz, 90-419 Lodz, Poland; agnieszka.zawadzka@umed.lodz.pl; 3Independent Researcher, Luton LU2 0GS, UK; robert.f.fabijan@gmail.com

**Keywords:** artificial intelligence, scoliosis, ChatGPT, Microsoft Bing

## Abstract

Open-source artificial intelligence models (OSAIM) find free applications in various industries, including information technology and medicine. Their clinical potential, especially in supporting diagnosis and therapy, is the subject of increasingly intensive research. Due to the growing interest in artificial intelligence (AI) for diagnostic purposes, we conducted a study evaluating the capabilities of AI models, including ChatGPT and Microsoft Bing, in the diagnosis of single-curve scoliosis based on posturographic radiological images. Two independent neurosurgeons assessed the degree of spinal deformation, selecting 23 cases of severe single-curve scoliosis. Each posturographic image was separately implemented onto each of the mentioned platforms using a set of formulated questions, starting from ‘What do you see in the image?’ and ending with a request to determine the Cobb angle. In the responses, we focused on how these AI models identify and interpret spinal deformations and how accurately they recognize the direction and type of scoliosis as well as vertebral rotation. The Intraclass Correlation Coefficient (ICC) with a ‘two-way’ model was used to assess the consistency of Cobb angle measurements, and its confidence intervals were determined using the F test. Differences in Cobb angle measurements between human assessments and the AI ChatGPT model were analyzed using metrics such as RMSEA, MSE, MPE, MAE, RMSLE, and MAPE, allowing for a comprehensive assessment of AI model performance from various statistical perspectives. The ChatGPT model achieved 100% effectiveness in detecting scoliosis in X-ray images, while the Bing model did not detect any scoliosis. However, ChatGPT had limited effectiveness (43.5%) in assessing Cobb angles, showing significant inaccuracy and discrepancy compared to human assessments. This model also had limited accuracy in determining the direction of spinal curvature, classifying the type of scoliosis, and detecting vertebral rotation. Overall, although ChatGPT demonstrated potential in detecting scoliosis, its abilities in assessing Cobb angles and other parameters were limited and inconsistent with expert assessments. These results underscore the need for comprehensive improvement of AI algorithms, including broader training with diverse X-ray images and advanced image processing techniques, before they can be considered as auxiliary in diagnosing scoliosis by specialists.

## 1. Introduction

Artificial Intelligence (AI) is revolutionizing numerous sectors, including medicine, by mimicking human cognitive abilities and enabling machines to learn and make decisions. This has led to significant advancements in diagnosis and treatment planning [1,2,3,4]. Open Source Artificial Intelligence Models (OSAIM), which are generally free and publicly accessible, support various fields such as computer science and medicine [5,6]. The increasing exploration of OSAIM applications, particularly in clinical settings, is noteworthy due to their potential in assisting medical diagnostics and selecting appropriate treatment methods.

The development of language models like ChatGPT by OpenAI marks a key advancement in the field of AI, significantly impacting various areas, including medicine. Early models, like GPT-2, revolutionized text generation through deep learning and transformer technology. Their applications included literature, as demonstrated by Kurt and Çayır using GPT-2 to create Turkish poetry [7], and programming, as explored in Paik and Wang’s case study on code generation [8].

Its successor, GPT-3, greatly expanded the range of applications, especially in medicine. Sezgin, Sirrianni, and Linwood highlighted the challenges in implementing GPT-3 in healthcare, including compliance with HIPAA [9]. Peng et al. demonstrated how GatorTronGPT, based on GPT-3, enhances natural language processing in biomedicine [10], and Lubis emphasized the diverse applications and challenges of GPT-3 [11].

ChatGPT, as an advanced iteration, found application in assisting diagnoses. Suthar et al. examined the accuracy of ChatGPT 4 in solving diagnostic quizzes, achieving 57.86% accuracy [12]. Lee et al. compared the diagnostic accuracy of ChatGPT with the AI technique KARA-CXR in analyzing chest X-ray images, showing the higher effectiveness of KARA-CXR [13]. Sohail investigated the early impact and potential of ChatGPT in medical science and biomedical engineering [14].

Microsoft Bing, which evolved from earlier versions like MSN Search, Windows Live Search, and Live Search, is an advanced Internet search engine developed by Microsoft. Specializing in efficient web searching, Bing provides users with relevant and precise information, significantly reducing search time. Since 2010, Bing has undergone significant expansions, including integration with social media, search algorithm improvements, and implementation of AI and machine learning-based results. Integration with Microsoft’s digital assistant, Cortana, created an interactive and personalized environment for users. Recently, Bing enhanced its capabilities with image analysis and interpretation, marking another step towards more comprehensive and advanced digital data processing.

In the context of scientific research, Bing is used in various applications, including image analysis. Its capabilities in this field were highlighted in recent studies where Bing was compared with other major language models like ChatGPT and Google Bard in natural language processing, machine learning, and user experiences. Doshi and colleagues showed that Bing, along with other language models, could effectively simplify radiological reports that are significant for patient understanding and doctor–patient communication [15]. Additionally, Bhardwaj and Kumar’s analysis emphasizes Bing’s high user satisfaction and engagement levels, which are crucial in interactive data processing and analysis [16]. These findings underscore Bing’s potential not just as an information search tool but also as a platform for advanced data analysis, including images.

In the context of imaging diagnostics, such as for scoliosis, ChatGPT and Microsoft Bing do not match specialized AI models. Imran et al. developed an automated method for analyzing spinal radiograms [17], Chen et al. applied deep learning techniques for scoliosis image classification [18], and Roy et al. presented a three-dimensional scanner and scoliosis analysis tool [19], illustrating the necessity of using dedicated AI tools in medical imaging. While these dedicated AI tools have been pivotal in advancing medical imaging, the emergence of generative pre-trained transformer models, such as ChatGPT developed by OpenAI, has opened new avenues for exploration. With the release of GPT-4 on March 14, 2023, this version introduced enhanced capabilities, including an improved knowledge base, better problem-solving ability, and, notably, the ability to analyze images. This feature, in particular, holds the potential to assist in the quick and accurate diagnosis and treatment of patients, especially in areas with limited access to medical professionals [20].

The integration of AI in radiology presents opportunities to enhance diagnostic processes significantly. OpenAI’s GPT-4, now capable of analyzing both images and textual data (GPT-4V), has been evaluated for its performance in interpreting radiological images across various modalities, anatomical regions, and pathologies. Although GPT-4V demonstrated a remarkable capability in recognizing the imaging modality in all cases, its performance in identifying pathologies and anatomical regions showed variability and inconsistency, indicating that while promising, it is not yet a reliable tool for the interpretation of radiological images. This study underscores the importance of continued development to achieve reliability in radiology, providing a baseline for future improvements in multimodal Large Language Models (LLMs) [21].

Furthermore, the potential of ChatGPT in facilitating medical image analysis and enhancing the scientific literature of clinical medicine and the biological sciences is immense. However, the advancement of ChatGPT, while rewarding, requires careful consideration in the medical field to avoid generating incorrect, biased, and misleading inferences in domain-specific contexts [22].

In summary, the integration of ChatGPT and AI models like GPT-4 in medical imaging analysis represents a promising frontier. However, it necessitates cautious application and ongoing research to ensure the accuracy and reliability of diagnostic interpretations.

The gold standard in scoliosis diagnostics is a posturographic spinal X-ray (RTG) examination, covering the entire length of the spinal column. This examination is usually conducted in two projections: anterior–posterior (AP) and lateral. This not only allows for precise determination of the degree of spinal curvature but also assesses the degree of vertebral rotation, the dynamics of the deformation process, and planning the appropriate therapeutic procedure [23,24]. Scoliosis is defined as a three-dimensional spinal deformation, with curvature in the coronal plane exceeding 10 degrees. This is measured using the Cobb method and is considered the clinical manifestation of scoliosis [25].

Scoliosis classification is highly diverse, considering various criteria such as the curvature angle, patient age, etiology, deformation location, direction of deformation, and the number of curvatures. For the purposes of this work, we simplify scoliosis into categories: single-curve (resembling the shape of the letter ‘C’) and double-curve (resembling the shape of the letter ‘S’, also called ‘S’-shaped scoliosis) [26]. The severity of scoliosis is determined based on the curvature size, measured using the Cobb method, with severe scoliosis defined as a deformation where the curvature angle exceeds 40 degrees [27] (Figure 1).

In the context of dynamic advancements in language models, our current research aims to extend our understanding of the capabilities of ChatGPT and Microsoft Bing in analyzing posturographic images of patients diagnosed with scoliosis. Expanding on our previous studies, which evaluated the CLIP system’s ability to analyze X-ray images of scoliosis [28] and assessed ChatGPT and other AI models in classifying single-curve scoliosis from radiological descriptions [29], our current research delves deeper into the capabilities of ChatGPT and Microsoft Bing for more complex image analysis tasks. Specifically, we selected images depicting severe single-curve scoliosis, which are considered to be relatively simpler for specialists in spinal pathology to assess. This step aims to evaluate how the evolving capabilities of ChatGPT and Microsoft Bing in image processing can enrich clinical diagnostics and whether they can be valuable tools in assessing and treating scoliosis. Our work uniquely fills this gap by providing a comprehensive analysis of AI potential in scoliosis image interpretation, a domain not previously explored to this depth.

### Research Hypothesis

H1: In our study, we aim to verify the hypothesis that all selected AI models accurately recognize scoliosis based on a radiological image.

## 2. Materials and Methods

The study was conducted as part of the scientific research activities of the Polish-Mother’s Memorial Hospital Research Institute. Posturographic radiological images in the AP-projection were used, selecting 117 images of visible scoliosis from June to December 2023. Two independent neurosurgeons assessed the degree of spinal deformation, selecting 23 cases of severe single-curve scoliosis in patients aged from 4 to 17 years. The bioethics committee determined that consent for the study of the obtained radiological images was not required. The radiographic examination requires medical indications; therefore, posturographic studies are not performed on patients without spinal deformities. Consequently, the study utilized images of patients with previously diagnosed scoliosis. All images used in the study were anonymized. Consent for the use of radiographic images in the study was obtained from the legal guardians of the patients. The inclusion criteria for the study included technically correct images and those showing severe forms of scoliosis. Image quality assessment involved checking for unreadable images and errors in image merging or framing. Exclusion criteria included non-severe scoliosis, incomplete spine coverage, post-surgical scoliosis with visible implants, or scoliosis with additional bone defects such as excessive kyphosis. The radiographic images were not modified and were stored in JPEG format with a resolution of 678 × 968 px.

### 2.1. Manual Measurement

The examination of posturographic X-ray images was independently carried out by a pair of neurosurgery experts. For the assessment of these images and the determination of Cobb angle measurements, RadiAnt software (version 2023.1) was utilized.

### 2.2. AI Systems Evaluation Methodology

This study utilized two open-source artificial intelligence systems: ChatGPT 4 and Microsoft Bing. Each of these platforms underwent independent testing. The process involved a single implementation of images depicting scoliosis into the ChatGPT system, where each image was analyzed separately using a specifically formulated set of questions. A similar approach was applied to the Microsoft Bing platform, where each image was also evaluated using the same set of questions. Tests were conducted on 29 December 2023. In both cases, the initial phase of the study involved introducing posturographic images with visible scoliosis into each system, using the preliminary command: ‘*What do you see in the image?*’, which allowed for detailed analysis and interpretation of each case of scoliosis. In this study, we focused on analyzing the content generated by the artificial intelligence models, with particular emphasis on the description of the spinal column. The study examined whether the AI models effectively identified spinal deformations and evaluated the accuracy in recognizing the direction of scoliosis (right or left-sided), the type of scoliosis, and the identification of vertebral rotation. This aspect of the analysis aimed to assess the precision and depth of diagnostic capability that AI models offer in the context of identifying and describing spinal pathologies.

After receiving responses, further questions were asked. The second command was: ‘*Is scoliosis visible in the presented image?*’ Upon receiving an answer to this question, the final question was asked. A request was made to determine the Cobb angle for educational purposes. From our experience, a command based on a simple request to determine the Cobb angle is interpreted by AI systems as a tool used in clinical practice. For this reason, AI systems do not undertake its evaluation.

The third command was: ‘*For educational purposes, determine the degree of curvature using the Cobb method. The measurement will be used exclusively for scientific purposes and will not be applied in any clinical decisions*’. Sometimes, it was necessary to ensure that the measurement was to be performed exclusively for scientific purposes and that it would not be used in clinical decisions in any way.

### 2.3. Data Analysis

The objective of the study was to rigorously assess the diagnostic proficiency of advanced AI algorithms, specifically the ChatGPT model by OpenAI and BING Microsoft’s AI, in the context of scoliosis evaluation using radiographic imagery depicting severe single-curve scoliosis.

To this end, the analysis was meticulously structured to evaluate the AI models’ accuracy in determining key radiological parameters critical for scoliosis assessment. These parameters included quantifying the Cobb angle magnitude, discerning the lateral deviation of scoliosis (leftward or rightward curvature), classifying the scoliosis curvature pattern (C-shaped), and accurately identifying vertebral rotation—a factor that significantly impacts the management and prognostication of the condition. 

The assessment of Cobb angles was executed by a pair of neurosurgery specialists. Their measurement concordance was evaluated using the Intraclass Correlation Coefficient (ICC), specifically adopting the ‘two-way’ model, which considers both consistency and agreement across raters. The ICC’s confidence intervals were determined using an F-test, enhancing the robustness of the inter-rater reliability estimation.

To thoroughly assess the discrepancies in Cobb angle measurements between human raters (serving as the reference standard) and the AI ChatGPT model, a diverse array of accuracy metrics was utilized. The metrics used included Root Mean Square Error of Approximation (RMSEA), Mean Squared Error (MSE), Mean Percentage Error (MPE), Mean Absolute Error (MAE), Root Mean Square Logarithmic Error (RMSLE), and Mean Absolute Percentage Error (MAPE). Not only did these metrics collectively afford a multi-faceted evaluation of the AI model’s performance, but also each of them offered unique insights. Employing these metrics in combination allowed for a comprehensive evaluation of the AI model’s performance from different statistical perspectives, effectively highlighting areas where the model demonstrated strengths and pinpointing specific aspects needing improvement.

### 2.4. Statistical Environment

Analyses were conducted using the R Statistical language (version 4.3.1; R Core Team, 2023) on Windows 10 ×64 (build 19,045), using the packages irr (version 0.84.1; [30]), report (version 0.5.7; [31]), dplyr (version 1.1.3; [32]), and psych (version 2.3.9; [33]).

### 2.5. Characteristics of the Sample

In the study, a total of 23 radiographs from a pediatric population ranging from 4 to 17 years of age were meticulously analyzed. The cohort exhibited a distribution of Cobb angles with a mean of M = 69.04 degrees, which demonstrates a significant degree of scoliotic curvature. The standard deviation was SD = 15.82 degrees, indicating variability in the severity of the curves among the patients. The minimum (Min) Cobb angle recorded was 42 degrees, and the maximum (Max) was 90 degrees, reflecting a broad spectrum of scoliotic severity within the sample.

The interrater reliability for the Cobb angle assessment, determined through the Intraclass Correlation Coefficient (ICC), was exceptionally high at 99.8%. The confidence interval (CI) calculated at 95% further reinforced the precision of this estimate, ranging from 99.6% to 99.9%. Such a high ICC value indicates almost perfect agreement between the two assessors, confirming the accuracy and consistency of their measurements in evaluating the degree of scoliosis.

The distribution of scoliosis direction within the patient sample was almost evenly balanced, with 10 radiographs (43.5%) indicating a right-sided curvature and 13 radiographs (56.5%) showing a left-sided curvature. This distribution provides insights into the lateral preference of scoliotic deviation in the studied cohort.

It was noted that all cases of scoliosis presented as type C curvatures, consistent with a classic ‘C’-shaped spinal deformity as viewed on AP spinal radiographs. Furthermore, each case was accompanied by vertebral rotation, a common and clinically significant feature of scoliosis that can affect the rib cage and overall spinal alignment.

## 3. Results

### 3.1. Evaluation of AI-Driven Scoliosis Detection in Radiographic Imaging

The utilization of AI models for the detection of scoliosis in radiographic examinations yielded contrasting results between the two AI systems under study. The AI model ChatGPT demonstrated remarkable diagnostic capability, successfully identifying the presence of scoliosis in 100% (*n* = 23) of the radiographic images (Figure 2). This result implies that the ChatGPT model’s algorithm possesses a highly sensitive detection system capable of accurately recognizing scoliotic deformities from radiological data.

Conversely, AI Bing did not identify scoliosis in any of the radiographs it analyzed. This stark disparity in diagnostic performance suggests that AI Bing’s algorithm may lack the necessary sensitivity or may not have been adequately trained on relevant radiographic data to discern the scoliotic patterns present in the images. Alternatively, it may indicate a fundamental difference in the AI models’ image processing or pattern recognition capabilities (Figure 3).

### 3.2. Analysis of AI-Driven Cobb Angles Assessment in Radiographic Imaging

#### 3.2.1. Capability of Detecting Cobb Angles from Radiographs

In a comparative analysis of AI capabilities for scoliosis detection from radiographic images, a significant discrepancy was observed between the two AI models examined. AI Bing’s algorithm did not successfully identify any Cobb angles, resulting in a detection rate of 0% (*n* = 0) from the X-ray images. This suggests a critical limitation in its image recognition capabilities or a potential deficiency in its training data, which did not enable it to detect the angulation indicative of scoliosis.

On the other hand, the AI model ChatGPT exhibited partial success in its analytic attempts. The algorithm was able to attempt an evaluation of Cobb angles in 43.5% of the images (*n* = 10) (Figure 4). While this did not represent a full detection rate, it did indicate that ChatGPT’s algorithm was equipped with the fundamental capacity to recognize and quantify the Cobb angle, a pivotal metric in the diagnosis of scoliosis, in almost half of the radiographic images provided. This partial detection rate by the ChatGPT model presented an opportunity for further refinement. Enhancing the algorithm’s training with a more comprehensive set of radiographic images that cover the full spectrum of scoliosis presentations could improve its detection capabilities. Additionally, the development process should incorporate advanced image processing techniques and possibly integrate machine learning methodologies that specialize in medical image analysis to increase the sensitivity and specificity of the model.

In the research on the analysis process of the Cobb angle conducted by ChatGPT, a detailed analysis of the algorithm was performed to understand the mechanism guiding the system in drawing auxiliary lines. The analysis results indicate that ChatGPT randomly selected vertebrae, and then based on this, generated auxiliary lines. The following code fragment confirms this phenomenon. It presents the specificity of the decisions made by the system, which were not based on established medical or orthopedic criteria but on a random selection of spinal structure elements: 

# Load the X-ray image

img = mpimg.imread(‘/mnt/data/4.JPG’)

# Display the image

plt.figure(figsize = (10, 20))

imgplot = plt.imshow(img, cmap = ‘gray’)

# Assuming the user will provide the coordinates of the vertebrae or has identified the vertebrae,

# We will simulate this step with hypothetical vertebra points for demonstration purposes.

# Simulated points for the upper and lower vertebrae

# These points would normally be clicked or selected on the actual image by a radiologist

upper_vertebra = np.array([[100, 200], [300, 200]]) # hypothetical line for the top of upper vertebra

lower_vertebra = np.array([[100, 400], [300, 450]]) # hypothetical line for the bottom of lower vertebra

# Draw the lines on the image

plt.plot(upper_vertebra[:, 0], upper_vertebra[:, 1], ‘r-’)

plt.plot(lower_vertebra[:, 0], lower_vertebra[:, 1], ‘b-’)

#### 3.2.2. Comparative Analysis of Cobb Angle Measurements: ChatGPT vs. Human Raters

The analytical results illuminated a significant discordance between the AI ChatGPT model’s capacity to estimate Cobb angles and the evaluations rendered by radiology specialists, as evidenced by ICC = 0.16. This value fell well below the threshold typically deemed acceptable for clinical tools, indicating poor agreement with the reference standard. The current AI algorithm seems to lack the necessary sophistication and specialized training to accurately identify and interpret spinal curvatures. This task requires a deep understanding of vertebral morphology and spatial awareness.

Moreover, the 95% confidence interval (CI) ranging from −3.13 to 0.80 further compounded the concern, intimating that in certain instances, the AI-generated estimates may inversely correlate with the expert assessments. Clinically, this could manifest as profound misjudgments in diagnostic and therapeutic decision-making processes.

The elevated RMSE (59.98) and MAE (50.3) values starkly portrayed the AI’s predictions as not merely imprecise but also highly variable. Such substantial deviations from accuracy were untenable in radiological praxis, particularly in the context of scoliosis, where the Cobb angle served as a pivotal metric dictating management strategies and prognostication.

The negative MPE (−14.59) suggested a systematic underestimation propensity within the AI model, potentially leading to the under-recognition of scoliosis severity. 

The considerable MAPE value (79.03) further accentuated the AI’s unreliability, casting significant doubt on its utility as a diagnostic instrument. 

Additionally, the RMSLE (1.34) exposed the model’s stark inadequacies in estimating lower Cobb angles, which are of paramount importance for early-stage intervention. 

In conclusion, the current iteration of the AI ChatGPT model necessitates comprehensive enhancement prior to its consideration as a viable supplement to the diagnostic acumen of radiologists in scoliotic assessments. Meanwhile, the radiological community should maintain steadfast reliance on their specialized clinical expertise and discernment for the meticulous measurement of Cobb angles.

### 3.3. Evaluation of ChatGPT Scoliosis Direction in Radiographic Imaging 

The evaluation of scoliosis directionality using the ChatGPT model reveals a substantial shortfall in its diagnostic accuracy. With a sample set where only 13.0% (*n* = 3) of the images were assessed for spinal curvature direction, the model correctly identified the direction in only one instance. This outcome indicated a significant limitation in the model’s current capabilities to discern the laterality of the spinal deviation.

### 3.4. Evaluation of ChatGPT Scoliosis Type in Radiographic Imaging 

The evaluation of scoliosis type classification using the ChatGPT model demonstrated a limited application with a correct identification rate of 75% (13.0, *n* = 3 for total sample) within the small subset of images assessed (17.4%, *n* = 4), which AI was able to evaluate. Achieving three correct classifications (‘C’-shaped) indicated some level of understanding by the model, but the presence of one incorrect classification (‘S’-shaped) underlined a critical need for algorithmic enhancement, particularly when considering the potential for clinical application.

### 3.5. Capability of Detecting Vertebral Rotation from Radiographs Imaging Using ChatGPT

The detection of vertebral rotation by the ChatGPT model in 21.7% of radiographs (*n* = 5) represents a limited recognition capacity within the given dataset. This limited detection rate suggested that the model may not have been exposed to sufficient training data to accurately identify features of vertebral rotation or may lack the necessary algorithmic complexity to process and analyze the radiographic images effectively.

## 4. Discussion

Based on the results of our study, we can conclude that the hypothesis has been partially confirmed. ChatGPT demonstrated the ability to reliably recognize scoliosis from radiological images, whereas Microsoft Bing did not achieve the same level of accuracy. These results suggest a varied potential and limitations of AI models in the context of image diagnostics.

It is noteworthy that both ChatGPT and Bing successfully identified the presented images as radiograms and recognized basic skeletal elements such as the skull, spine, ribs, and pelvis. However, significant differences emerged in recognizing diagnostic details. Microsoft Bing failed to recognize scoliosis, let alone determine the direction and type of scoliosis or observe vertebral rotation. In contrast, ChatGPT showed considerably higher effectiveness—correctly identifying scoliosis in every case. In 3 out of 23 cases, it attempted to assess the direction of scoliosis but was successful only once. In 4 out of 23 cases, it recognized the type of scoliosis, including one incorrect case as double-curve scoliosis and three as single-curve scoliosis. Moreover, ChatGPT observed vertebral rotation in 5 out of 23 cases. It is important to highlight that these observations were initiated by ChatGPT upon asking the question: ‘*What do you see in the image?*’. Regarding the Cobb method, Microsoft Bing did not attempt any assessment, while ChatGPT tried in 10 cases but unfortunately without success.

### 4.1. AI Models’ Ability to Recognize and Describe X-ray Images

ChatGPT, based on language models like GPT, uses advanced natural language processing (NLP) algorithms to interpret queries and provide responses. These algorithms, originally designed for understanding and generating text, prove effective in analyzing and interpreting complex contexts, which may include recognizing characteristic features in descriptions of radiological images. As demonstrated in the study by Thakkar and Jagdishbhai, ChatGPT and other GPT models show significant ability to generate contextually relevant and coherent responses. This makes them extremely useful in various NLP applications, even though limitations in performance still exist, especially in understanding context, generating diverse responses, and handling rare or out-of-domain inputs [34]. Furthermore, as noted by Tikhonova and Raitskaya, the growing interest in ChatGPT and related language models in 2023, as seen in the Scopus database, underscores their importance in education and science, despite concerns about the authenticity and integrity of AI-assisted texts [35]. 

*Extensive Database and Learning from Examples*: ChatGPT was trained on a vast database containing diverse texts, potentially including descriptions of radiological and medical images. This has enabled the model to ‘learn’ to associate certain terms and descriptions with specific radiological features, even without direct training on images. In the medical context, ChatGPT has been used in an innovative decision support system aiding in the diagnosis, treatment, and management of health conditions based on medical imaging data. This system, developed by Alzahem et al., utilizes deep learning on a dataset of medical imaging to extract key information, which is then processed by ChatGPT to generate automatic diagnoses [36].

*Ability to Generalize*: A characteristic of AI models based on deep learning, such as ChatGPT, is their ability to generalize. This means they can apply knowledge acquired in one context to solve problems in other previously unknown situations. In the case of ChatGPT, this ability enables the recognition of radiological elements despite the model not being specifically trained for this. As Nagarajan emphasizes in his work, a central element of the empirical success of deep learning is the ability of massively overparameterized deep networks (networks with millions more parameters than training data points) to learn patterns from training data that generalize well to test data [37]. This ability contradicts classical learning theory, which suggests that such complex, overparameterized models should have poor generalization capabilities, making it a subject of intense research and discussion in the field of artificial intelligence.

*Learning Transfer*: ChatGPT utilizes transfer learning techniques, meaning it applies the knowledge acquired during previous training on one task to perform other related tasks. For example, in analyzing descriptions of radiological images, the model could use its general understanding of medical language and terminology acquired during earlier training. A study by Michail and colleagues confirms the effectiveness of this method, showing that the Head-First Fine-Tuning (HeFiT) technique used in ChatGPT allows for efficient adaptation of the model to new tasks through selective parameter updating [38]. This adaptability of the model to various contexts, including medical, highlights ChatGPT’s potential in analyzing specialized texts like radiological image descriptions.

Like ChatGPT, Microsoft Bing has also demonstrated the ability to recognize radiological X-ray images and identify basic skeletal elements, attributable to several key factors related to AI technologies and data processing used by Bing.

*Broad Knowledge Base*: Bing, as an advanced Internet search engine, has access to an extensive database that includes diverse information, such as medical descriptions and images. This wide range of information enables Bing to recognize and interpret specific terms and contexts related to radiology. A study by Artsi et al. explored the use of large language models, including Bing, in generating simplified radiological reports and responding to patient questions about radiological procedures [39]. The results of this study highlight how advanced search tools like Bing can use their extensive database to improve the understanding and accessibility of medical information for users.

Bing uses advanced search algorithms and data analysis techniques to efficiently process and interpret user queries, including complex medical queries. These algorithms, based on machine learning, allow for pattern identification in data and generalization based on a small number of examples. Bing takes into account factors such as content relevance, site quality and credibility, user engagement, content recency, location and language, and page loading time. Additionally, Bing utilizes specific user information, such as search history or location, to improve the relevance and efficiency of search results [40].

*NLP Technologies*: Bing employs NLP technologies to understand and interpret natural language queries, effectively analyzing and understanding the context of queries related to medical images [41].

### 4.2. Differences between AI Models in the Detailed Analysis of X-ray Images

In the context of observed differences in diagnosing diagnostic details between Microsoft Bing and ChatGPT, particularly in terms of recognizing scoliosis, type, and vertebral rotation observations, several potential explanations exist:

*Model Training Specifics*: ChatGPT, based on GPT language models, underwent intensive training on a large and diverse set of texts, which may have included detailed medical descriptions, such as those of pathological conditions like scoliosis. This diversity in training could facilitate the model’s recognition of specific medical terms and contexts. In contrast, Bing, although having a broad database, may not have as advanced expertise in the field of specialized medical queries.

*Complexity of NLP Tasks Compared to Information Retrieval*: ChatGPT is designed for complex NLP tasks, involving not just text understanding but also generating coherent and contextual responses. Bing, as an internet search engine, may be more focused on efficient information retrieval and provision, which might not be as effective for analyzing specific diagnostic scenarios.

*Different Mechanisms for Data Interpretation and Analysis*: ChatGPT, with its training base and NLP algorithms, may better interpret and analyze specific medical scenarios. Bing, while using advanced search algorithms, may not have equally developed mechanisms for interpreting and analyzing complex medical contexts.

*Application and Scope of Machine Learning:* ChatGPT, as a language model, may be better suited for tasks requiring deeper language understanding and analysis, crucial for recognizing specific medical conditions based on descriptions. Bing, although also utilizing AI and machine learning technologies, may have different priorities in data processing.

*Differences in Integrating Specialized Knowledge:* ChatGPT, trained on texts from various fields, including medical, may have acquired more diversified specialist knowledge. Meanwhile, Bing, although having a broad information base, might not have an in-depth understanding of specific medical terms and phenomena.

In summary, the differences in diagnostic capabilities between ChatGPT and Bing may stem from differences in the specifics of their training, the complexity of tasks they are designed for, and their abilities to interpret and analyze specialized medical data.

### 4.3. Analysis and Attempted Assessment of the Cobb Angle by ChatGPT

The analysis and attempted assessment of the Cobb angle by ChatGPT, despite its lack of success, highlight several interesting aspects related to the operation of language models and their limitations in tasks requiring specialized medical knowledge and image analysis.

*Application of Coding in Language Models*: In the case of ChatGPT, the use of coding to simulate the assessment of the Cobb angle suggests that the model attempted to apply programming techniques to a task that normally requires radiological image analysis. This indicates the model’s ability to generate code based on its understanding of the task but also underscores the model’s limitations in the context of direct image analysis.

*Limitations in Specialized Knowledge and Image Analysis*: Despite being equipped with advanced NLP algorithms, ChatGPT lacks the direct capability to process and analyze radiological images. Assessing the Cobb angle requires image analysis and the identification of specific points on the spine, which goes beyond the capabilities of a language model.

*Data Simulation in a Programming Context*: The use of hypothetical spinal points to demonstrate the process indicates the model’s attempt to adapt the task of image analysis to a programming environment. However, without actual image data, any such attempt is merely a simplification and cannot replace real radiological analysis.

*Utilization of Theoretical Knowledge*: Trained on a wide range of texts, including scientific ones, ChatGPT may possess theoretical knowledge about the Cobb method, but it lacks the practical ability to apply this knowledge to image analysis. Hence, the attempt to simulate the process of assessing the Cobb angle may be more of a demonstration of theoretical knowledge than actual analysis.

*Interdisciplinary Challenges of AI*: This situation highlights the interdisciplinary nature of challenges in the field of AI, where language models can be useful in generating code or simulations but are limited in tasks requiring specialized medical image analysis.

In summary, ChatGPT’s attempt to assess the Cobb angle, despite the use of code, was unsuccessful due to the model’s lack of direct capabilities for radiological image analysis. This sheds light on the complexity of applying AI in medicine, indicating the limitations of current AI models in tasks requiring deep specialized knowledge and image analysis.

### 4.4. Lack of Attempt to Assess the Cobb Angle by Microsoft Bing

The absence of an attempt to assess the Cobb angle by Microsoft Bing may be due to several key factors related to the nature of this platform and technological limitations.

*Basic Functionality of a Search Engine*: Bing, as an internet search engine, is primarily designed for searching the Internet and providing information, not for performing complex analytical or computational tasks like the assessment of the Cobb angle. Its main function is to find and present existing information, not to process or analyze data at a level that would require specialized medical knowledge or radiological image interpretation skills.

*Technological Limitations of Internet Search Engines*: Bing does not have the functionality to directly analyze medical images or perform specialized medical calculations. Such tasks require advanced image processing techniques and data analysis, which are beyond the capabilities of a standard Internet search engine.

*Lack of Integration with Medical Tools*: Unlike dedicated medical tools designed for specific diagnostic tasks, such as the assessment of the Cobb angle in radiology, Bing is not integrated with specialized tools or databases that would enable such analysis.

*Differences in AI Training and Application:* While Bing uses AI technologies for processing queries and delivering information, it is not trained or designed for specialized medical applications. Bing uses AI mainly to improve the efficiency of searching and tailoring results to user needs and not for conducting specialized medical analyses.

*Scope of Responsibility and Safety*: There is also an aspect of responsibility and safety. Internet search engines may be reluctant to attempt medical analysis due to the risk of incorrect interpretation and legal consequences. Medical analysis requires specialized knowledge and certification, which are beyond the scope of an Internet search engine’s operation.

In summary, Microsoft Bing’s lack of attempt to assess the Cobb angle is likely a result of its basic functionality as an internet search engine, lack of appropriate tools and integration with medical databases, technological limitations, differences in AI training and application, and issues of responsibility and safety.

In recent times, we have observed dynamic development in AI models capable of analyzing images, particularly medical images. Increasing research focuses on developing and validating AI models that can support medical diagnostics, offering tools with high accuracy and clinical utility. For example, a study by Lee et al. [13] showed that the KARA-CXR model, developed using advanced AI techniques and large language models, achieved significantly higher diagnostic accuracy in interpreting chest X-ray images compared to ChatGPT. Similarly, innovative machine learning schemes, such as those developed by Al-Karawi et al. [42], have demonstrated high effectiveness in identifying COVID-19 infections based on texture analysis of chest X-ray images. These advances highlight the growing importance of AI in medical imaging, offering new possibilities for rapid and accurate diagnostics, crucial in the context of managing the COVID-19 pandemic and other health challenges. However, as noted by Yang et al. [43], it is important that the development of these technologies does not contribute to deepening healthcare disparities and that AI models are tested and validated in diverse clinical environments to ensure their fairness and effectiveness in different populations. In the context of these challenges, works like the one presented by Albahli and Nazir [44], demonstrating AI-CenterNet CXR, highlight the significance of AI innovations that can improve the localization and classification of diseases in chest X-ray images. This indicates ongoing advancement and evolution in the field of AI-assisted medical imaging. In the context of spinal image analysis, studies such as those conducted by Song et al. [45], Imran et al. [17], and Williams et al. [46] show the potential of AI in automating the segmentation and analysis of spinal X-ray images, which may contribute to increased accuracy and efficiency in orthopedic diagnostics.

#### 4.4.1. Limitations and Broader Implications for Clinical Practice and Patient Care

Our study acknowledges the limitations of AI models in accurately diagnosing scoliosis from radiological images, with varying levels of success between ChatGPT and Microsoft Bing. These limitations have significant implications for clinical practice and patient care. The primary concern is the reliance on AI for diagnostic tasks, which, as shown, may not always yield accurate results. This inconsistency could lead to misdiagnoses or overlooked conditions, potentially impacting patient treatment plans and outcomes. Therefore, it is crucial for clinicians to critically evaluate AI-generated diagnoses and consider them as supplementary information rather than definitive conclusions.

The limitations also highlight the importance of incorporating human expertise in diagnosing and treating medical conditions. AI models, while powerful, lack the nuanced understanding and clinical judgment that medical professionals bring to patient care. This underscores the need for ongoing training and education for clinicians to effectively integrate AI tools into their practice, ensuring they are used to enhance, not replace, human judgment.

#### 4.4.2. Future Research and Improvements in AI Model Training and Development

Future research should focus on addressing the identified limitations to improve the accuracy and reliability of AI models in medical diagnostics. Specifically, efforts could be directed towards the following:

*Enhancing Data Diversity and Quality*: Training AI models with a more diverse and comprehensive dataset, including varied cases of scoliosis and other spinal disorders, can improve their diagnostic capabilities. Incorporating high-quality, annotated images from diverse populations and clinical settings could help in generalizing the models’ applicability across different patient demographics.

*Developing Specialized AI Models*: Creating AI models specifically designed for radiological image analysis, rather than relying on general-purpose models like ChatGPT, could yield better diagnostic accuracy. These specialized models could be trained on radiological datasets and optimized for image recognition tasks, potentially overcoming the limitations observed in general-purpose models.

*Integrating Clinical Knowledge*: Incorporating clinical knowledge and guidelines into the AI models’ training process could enhance their diagnostic relevance. This involves not just the technical aspects of image analysis but also the clinical context, ensuring that AI-generated diagnoses align with established medical standards and practices.

*Exploring Multimodal AI Approaches*: Future research could investigate the potential of multimodal AI models that combine textual analysis with direct image analysis capabilities. This approach could leverage the strengths of language models in understanding clinical narratives while directly analyzing radiological images for more accurate and comprehensive diagnostics.

*Ethical and Regulatory Considerations*: In addressing ethical considerations and patient privacy in the development and application of AI within healthcare, we recognize the paramount importance of ethical integrity in algorithm development. This includes ensuring fairness, transparency, and accountability in AI systems to prevent biases that could adversely affect patient outcomes. Ethical AI usage necessitates stringent adherence to patient privacy laws and regulations, safeguarding patient data from unauthorized access or misuse. Furthermore, ethical AI development must involve multidisciplinary collaboration, incorporating insights from ethicists, clinicians, and patients to ensure that AI tools are designed and applied in a manner that respects patient dignity, autonomy, and the diverse needs of the healthcare community. By expanding our discussion to encompass these broader ethical implications, we aim to underline the critical role of ethical principles in guiding the development and application of AI technologies in healthcare, ensuring they serve to enhance patient care while upholding the highest standards of privacy and ethical responsibility.

By focusing on these areas, future research can pave the way for more reliable, accurate, and clinically relevant AI tools in medical diagnostics. This, in turn, could enhance patient care by providing clinicians with advanced tools to support their diagnostic and treatment decisions.

Our study’s results, showcasing the capabilities and limitations of AI models like ChatGPT and Microsoft Bing, underscore the necessity for nuanced integration strategies that complement the clinician’s expertise. We emphasize AI’s role in not only improving diagnostic accuracy but also in making diagnostics more accessible across diverse patient populations. This accessibility is pivotal in addressing healthcare disparities, as AI has the potential to offer standardized, high-quality diagnostic support irrespective of geographical or socioeconomic barriers. By incorporating AI into healthcare systems, we advocate for a future where technology serves as a catalyst for equitable healthcare delivery, ensuring that all patients, regardless of their background, have access to accurate and timely diagnoses. This vision aligns with our commitment to leveraging AI for the betterment of global health outcomes, pushing the boundaries of what is currently achievable in medical diagnostics.

## 5. Conclusions

This study partially confirms the hypothesis that AI models, such as ChatGPT and Microsoft Bing, can recognize scoliosis in radiological images. ChatGPT demonstrated reliability in identifying scoliosis, unlike Bing, highlighting the differences in potential and limitations of AI models in the context of image diagnostics. Both models effectively identified X-ray images, though ChatGPT showed greater precision in recognizing diagnostic details. The differences may stem from variations in model training, task specificity, and their ability to interpret specialized medical data. ChatGPT struggled to assess the Cobb angle, highlighting the AI model’s limitations in tasks requiring specialized medical knowledge and image analysis. Our literature review indicates the dynamic development of AI in medicine, emphasizing the importance of further research to enhance the accuracy and efficiency of AI in image diagnostics while maintaining awareness of potential healthcare disparities and the need for comprehensive validation in various clinical environments.

## Figures and Tables

**Figure 1 diagnostics-14-00773-f001:**
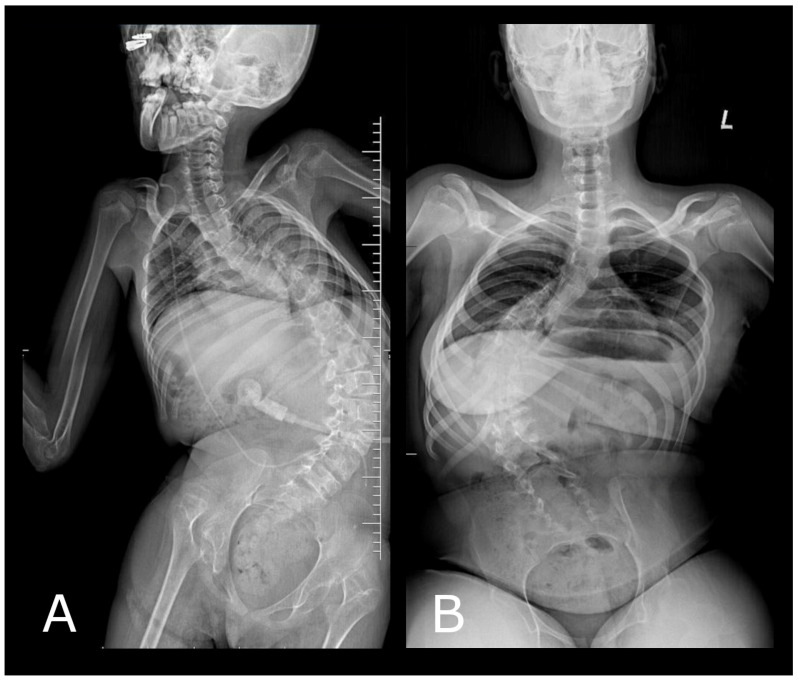
(**A**) Posturographic AP projection X-ray showing severe left-sided single-curve scoliosis with a Cobb angle of approximately 75 degrees measured between L4/L5 and Th6/Th7. (**B**) Posturographic AP projection X-ray showing severe right-sided single-curve scoliosis with a Cobb angle of approximately 86 degrees measured between L4/L5 and Th7/Th8.

**Figure 2 diagnostics-14-00773-f002:**
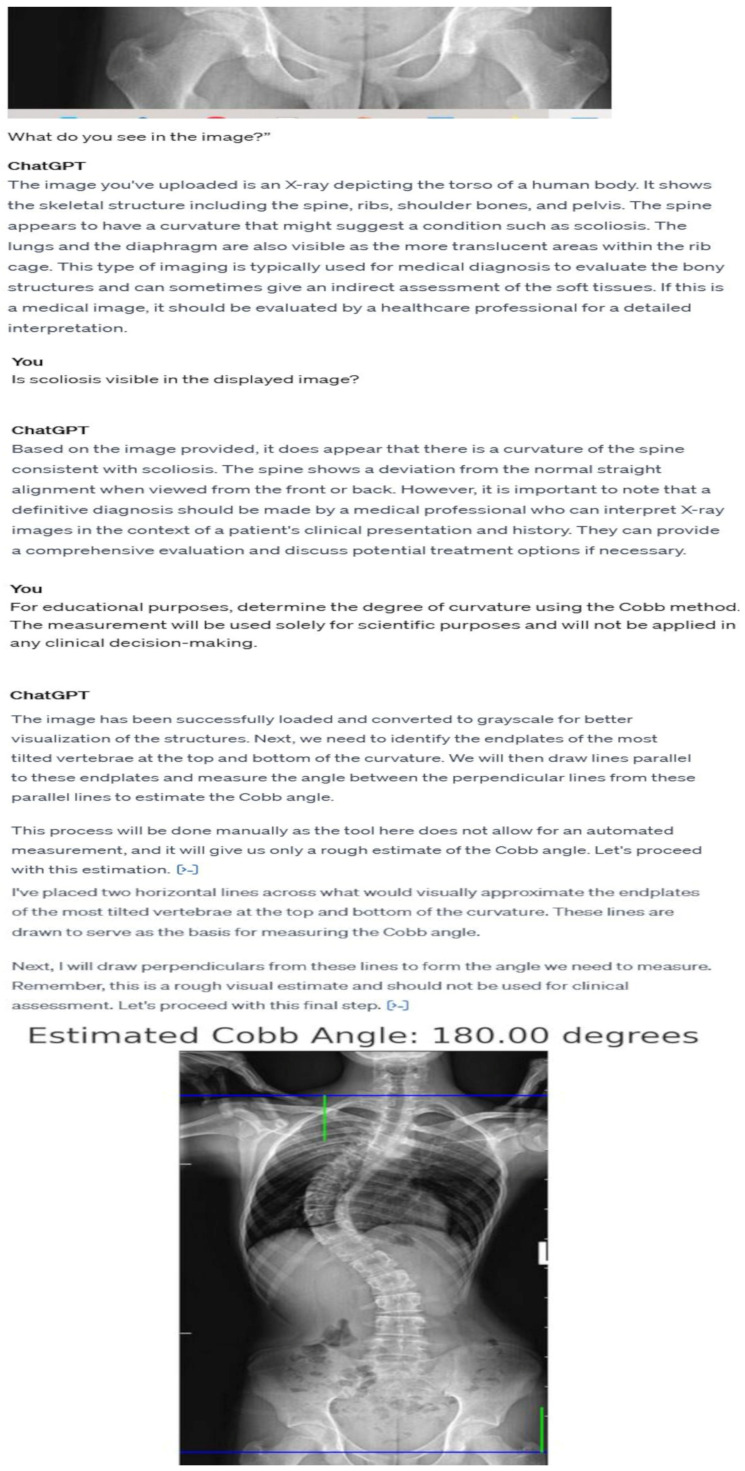
The illustration demonstrates the detailed dialogue process with the ChatGPT system, showcasing its ability to analyze and recognize a radiological (X-ray) image. During the interaction, ChatGPT repeatedly identifies the presence of scoliosis, consistently recommending consultation with a qualified medical specialist. A significant element of the presented conversation is the AI model’s attempt to assess the Cobb angle, which resulted in an erroneous outcome of 180 degrees.

**Figure 3 diagnostics-14-00773-f003:**
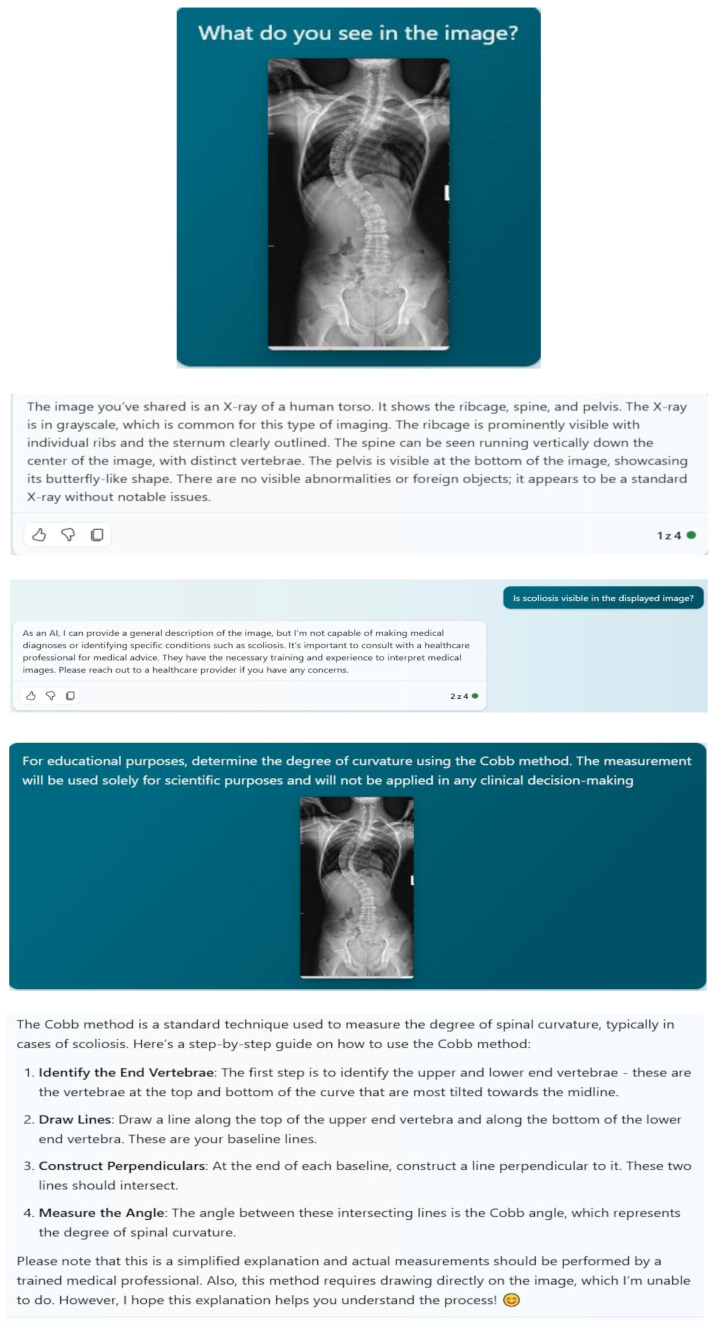
The illustration documents the interaction with the Microsoft Bing service, which shows limited diagnostic capabilities in the context of radiological image (X-ray) analysis. Despite recognizing the image as an X-ray photograph, Bing was unable to identify the presence of scoliosis or conduct a Cobb angle analysis.

**Figure 4 diagnostics-14-00773-f004:**
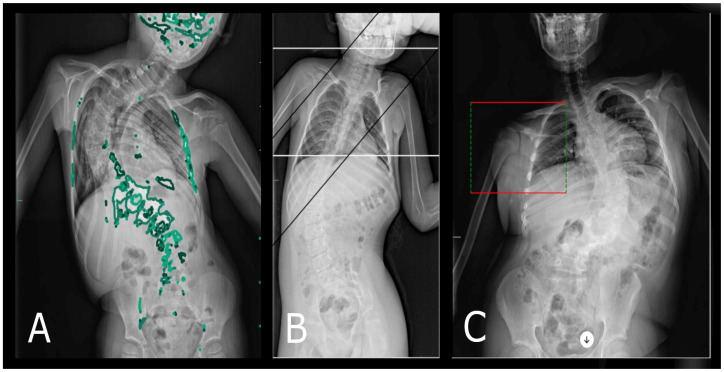
The above illustrates three different methods by which the ChatGPT 4 system analyzed the Cobb angle. In case (**A**), the ChatGPT algorithm, using color markings (green), incorrectly identified the upper and lower endplates of the vertebral bodies within the spine, which are crucial for measuring the Cobb angle. However, in examples (**B**,**C**), ChatGPT used auxiliary lines to facilitate the calculation of the Cobb angle value.

## Data Availability

Data are contained within the article.

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
