# Peer review of "Artificial Intelligence in Medical Imaging: Analyzing the Performance of ChatGPT and Microsoft Bing in Scoliosis Detection and Cobb Angle Assessment"

_diagnostics, 2024, doi:10.3390/diagnostics14070773_

Round 1

Reviewer 1 Report

Comments and Suggestions for Authors

1.This article is not innovative enough.

2.There are many AI models, why choose chatGPT and Microsoft Bing?

3.A standard should be established to evaluate the results of AI models

4.How to improve the AI ​​model should be the focus of this article.

Reviewer 2 Report

Comments and Suggestions for Authors

The presented study evaluating the capabilities of AI models, including ChatGPT and Microsoft Bing, in the diagnosis of single-curve scoliosis based on posturographic radiological images.

-The topic is very interested, but the paper can be improved by utilizing other radiological image based medical applications. To assest the powerful of nowadays artificial intelegence tools, I suggest to include different area of medical diagnosis based images. 

-The evaluation metrics RMSEA, MSE, MPE, MAE, RMSLE, and MAPE are not enough to assess the performance. 

-The AI results should be compared with the state of the art results presented in the literature review. Also, just suggestions are presented to improve the AI performance by increasing the training data and so on, however , real  work should be performed in this direction.

Reviewer 3 Report

Comments and Suggestions for Authors

This paper presents a significant contribution to the field of AI in medical diagnostics, specifically in the detection and assessment of scoliosis through radiographic imaging using ChatGPT and Microsoft Bing. Your work provides valuable insights into the capabilities and limitations of these AI models, offering a foundation for future research and development. However, there are several areas where further clarification, improvement, and expansion could enhance the impact and relevance of your study.

  1. Technical Detailing and Reproducibility: While the methodology section provides a good overview of the procedures and analyses undertaken, greater detail could be beneficial. Specifically, elaborating on the criteria for image selection and the rationale behind the specific questions used to evaluate the AI systems would enhance the reproducibility and depth of your study. Consider including more information on the algorithmic challenges encountered and the potential solutions explored.

  2. Comparison with Existing Literature: The introduction does a commendable job of situating your research within the broader AI and medical diagnostics landscape. However, a more thorough literature review, particularly focusing on previous applications of ChatGPT and Microsoft Bing in medical diagnostics, could strengthen the paper. Highlighting how your work builds upon or diverges from existing studies would provide clearer context for your contributions.

  3. Statistical Analysis and Methodological Rigor: Your statistical analysis is robust, yet further discussion on the choice of metrics and their implications for clinical practice would be beneficial. Additionally, considering alternative statistical models or validation techniques could offer deeper insights into the AI models' performance and reliability.

  4. Discussion of Limitations and Future Work: While the paper acknowledges the limitations of the AI models tested, a more comprehensive discussion on the broader implications of these limitations for clinical practice and patient care is warranted. Additionally, outlining specific areas for future research and potential improvements in AI model training and development could guide subsequent work in this field.

  5. Language and Clarity: The paper is generally well-written, but minor grammatical errors and awkward phrasings detract from its overall clarity. A thorough proofreading and editing pass would enhance the readability and professional quality of your manuscript.

  6. Ethical Considerations and Patient Privacy: Your ethical considerations regarding patient privacy and data use are commendable. However, expanding on the implications of AI in healthcare, particularly regarding ethical considerations in algorithm development and application, would add depth to your discussion.

  7. Broader Implications and Clinical Relevance: Finally, a more detailed exploration of the potential clinical applications of your findings, including how AI could be integrated into existing diagnostic workflows, would be valuable. Discussing the potential for AI to address healthcare disparities and improve diagnostic accuracy across diverse patient populations could broaden the impact of your work.

Your study represents a valuable step forward in understanding AI's potential and limitations in medical imaging diagnostics. By addressing these suggestions, your paper could make a more significant contribution to the field, guiding future research and clinical applications of AI in healthcare.

Comments on the Quality of English Language

The quality of English language used in the manuscript is generally good, providing a clear and coherent presentation of the research and its findings. However, there are areas where improvements could significantly enhance the readability and professional quality of the paper.

  1. There are instances of grammatical errors and awkward phrasing that disrupt the flow of the text. Attention to subject-verb agreement, proper use of articles, and avoidance of run-on sentences would improve the overall grammatical quality.

  2. Ensuring consistency in the use of technical terms and acronyms throughout the paper is crucial for readability and clarity. Occasionally, terms are introduced without definition or used interchangeably without clarification. A consistent and clear approach to terminology will aid in the reader's comprehension.

  3. Some sentences are overly complex or contain multiple ideas, making them difficult to follow. Simplifying these sentences and focusing on one idea at a time can enhance clarity and ensure that the readers easily grasp the key points being made.

  4. It is evident that further proofreading and editing are needed to correct typographical errors and improve sentence structure. These revisions are essential for maintaining a professional standard of writing and ensuring that the paper accurately conveys the intended meaning without distraction.

  5. The paper frequently employs the passive voice, which, while common in scientific writing, can sometimes make sentences less engaging or direct than they might otherwise be. Consider balancing the use of passive and active voice to make the text more dynamic and easier to read, especially in sections describing the research process and findings.

  6. While the paper generally does a good job of explaining complex AI and medical concepts, there are moments where further simplification or additional explanations could be beneficial. Ensuring that complex ideas are accessible to a broader audience, including those who may not be specialists in AI or medical imaging, will increase the paper's impact.

In summary, focusing on these areas of improvement for the English language quality will not only enhance the readability and accessibility of your manuscript but also contribute to its overall credibility and professionalism. A meticulous approach to language, grammar, and clarity will ensure that your research is presented in the best possible light, facilitating a wider understanding and appreciation of your important work.

Round 2

Reviewer 2 Report

Comments and Suggestions for Authors

No comments

Author Response

I sincerely appreciate your thorough review and am grateful for your recognition of the revisions made to our publication.